# Telomere Length, a New Biomarker of Male (in)Fertility? A Systematic Review of the Literature

**DOI:** 10.3390/genes14020425

**Published:** 2023-02-07

**Authors:** Anne-Julie Fattet, Maxime Chaillot, Isabelle Koscinski

**Affiliations:** 1Centre d’AMP Majorelle-Atoutbio, 95 Rue Ambroise Paré, 54000 Nancy, France; 2Service de Médecine et Biologie du Développement et de la Reproduction, 38 Boulevard Jean Monnet, 44000 Nantes, France; 3Faculté de Médecine, Université de Nantes, 44000 Nantes, France; 4Inserm U1256, Nutrition Genetics Environmental Risks Exposure (NGERE), Université de Lorraine, 54000 Nancy, France; 5Centre d’AMP Hôpital Saint Joseph, 26 Bd de Louvain, 13008 Marseille, France

**Keywords:** telomere length, telomerase, male infertility, sperm, semen parameters

## Abstract

Male factors are suspected in around half cases of infertility, of which up to 40% are diagnosed as idiopathic. In the context of a continuously increased resort to ART and increased decline of semen parameters, it is of greatest interest to evaluate an additional potential biomarker of sperm quality. According to PRISMA guidelines, this systematic review of the literature selected studies evaluating telomere length in sperm and/or in leukocytes as a potential male fertility biomarker. Twenty-two publications (3168 participants) were included in this review of experimental evidence. For each study, authors determined if there was a correlation between telomere length and semen parameters or fertility outcomes. Of the 13 studies concerning sperm telomere length (STL) and semen parameters, ten found an association between short STL and altered parameters. Concerning the impact of STL on ART results, the data are conflicting. However, eight of the 13 included studies about fertility found significantly longer sperm telomeres in fertile men than in infertile men. In leukocytes, the seven studies reported conflicting findings. Shorter sperm telomeres appear to be associated with altered semen parameters or male infertility. Telomere length may be considered as a new molecular marker of spermatogenesis and sperm quality, and thus is related to male fertility potential. However, additional studies are needed to define the place of the STL in the assessment of individual fertility.

## 1. Introduction

### 1.1. Background

Over the past decades, the occurrence of male infertility has continuously increased and represents 50% of all infertility cases worldwide. But in almost half of these cases, the underlying cause of sperm alteration remains unexplained and is reported as idiopathic [1]. 

Latent causes of these cases are multifactorial, including genetic, lifestyle and environmental factors and often a combination of them [2]. Thus, questions concerning the molecular mechanisms underlying male infertility are still being raised. 

The evaluation of male [in]fertility is usually based on semen analysis, i.e., sperm count with a mobility and morphology assessment. Sperm abnormalities refer to structural and/or functional defects in sperm, whose frequency is higher in infertile men. But this basic examination of sperm parameters sometimes fails to detect any abnormality and clearly distinguish fertile from infertile men [3]. Many tests have been proposed to diagnose male infertility in addition to traditional semen parameters to better evaluate sperm quality and function. A huge interest is the use of these tests in men undergoing infertility with normal standard semen parameters, in order to identify new diagnostic and prognostic biomarkers of sperm function and quality. In recent years, many studies reported that telomeric instability may affect gametes quality and eventually fertility. Telomere homeostasis has been gaining importance and may pinpoint a novel biomarker of human infertility [4].

The term “telomere” is a Greek-derived word which means “end” (“*telos*”) and “part” (“*meros*”). Telomeres were first described by Hermann Muller 80 years ago, during his works on *Drosophila*. They consist of specialized non-coding nucleoprotein structures localized at the ends of eukaryotic chromosomes [5]. These DNA–protein complexes are repetitive guanine rich TTAGGG sequences associated with numerous proteins known as the shelterin complex [6]. These heterochromatic structures are essential during cell division to maintain genomic integrity by protecting chromosomes’ ends, to be identified as DNA double-stranded breaks, which would trigger DNA damage response (DDR) or chromosomes’ end-to-end fusions [5]. During each replication round, the complete replication of the 3′ terminus of the DNA cannot be achieved by DNA polymerase, and this leads to terminal erosion of chromosomes; this phenomenon is called the “end replication problem”. The presence of telomeres enables progressive shortening of chromosomes’ extremity without inducing genetic information loss, which maintains genomic stability [7]. The progressive attrition leads to a critically short telomere length (TL) which induces proliferation arrest, senescence or apoptosis of somatic cells [8]. Thus, in each tissue, the stem cell TL at a given age is the result of the initial length (at birth) and the shortening due to successive mitoses.

Telomerase is a reverse transcriptase compounded of two subunits: the catalytic one, called “telomerase reverse transcriptase” (TERT), and the human RNA matrix, called the “telomerase RNA component” (TERC). This ribonucleoprotein is able to limit telomere shortening by extending the guanine-rich sequence and thus slowing down the continuous loss of DNA at each cell division [9]. Almost all somatic cells have no telomerase activity contrary to stem, embryonic and germs cells. Since telomerase prevents telomere erosion, this enzyme allows cells to enhance their proliferative capacity. Telomerase is active in most cancer cells, allowing theoretically unlimited proliferation [10].

In adult humans, mean TL in somatic cells ranges from 4 to 12 kb and from 6 to 20 kb in spermatozoon, but these values differ greatly within an individual and between spermatozoon from the same ejaculate [11,12,13]. In male germ cells, TL is preserved and increased during spermatogenesis thanks to telomerase [5,13]. Telomerase is present in testes from fetal to adult life. Evaluated at each meiotic stage, the telomerase activity was highest in spermatogonia, decreasing in spermatocytes and spermatids. However, in spermatozoon, no telomerase activity was measured [13,14]. Although it is well established that telomere length decreases with time in somatic cells, telomere dynamics in sperm cells appear to be different. Some authors described a telomere elongation of almost 135 bp per year according to Baird et al. [15]. This length increase is due to the telomerase and explains why sperm telomeres are generally longer than somatic cells [16]. All these facts suggest an important role of TL in male gametogenesis and fertility [5].

### 1.2. Rationale of Systematic Review of the Literature

Deciphering the molecular mechanisms underlying telomere function and how their disruption can have consequences on sperm parameters is therefore a huge challenge. These discoveries may be crucial in understanding the extent to which sperm participate in the proliferative properties of embryonic cells and may open up new methods of treatment for male infertility. Despite the fact that the importance of sperm telomeres is not completely known, recent studies reported a potential role in male (in)fertility [17]. This evidence suggests that sperm telomere length (STL) could be positively or negatively associated with male fertility, and that it might be a potential new biomarker of sperm quality. 

Moreover, several authors have investigated the possible link between TL in somatic cells, such as leukocyte telomere length (LTL), and sperm parameters and/or male fertility. The objective would be to dispose of a marker available from a simple blood test. The hypothesized link between LTL and male fertility is based on the hypothesis that LTL and STL might be positively correlated [18,19,20], since the length of the telomeres in given cells depends on (1) the telomere length at birth in the stem cells at the origin of the given cells, (2) the mitotic index of the tissue, (3) the age of the individual, (4) the telomerase activity in the given tissue, and (5) all pathophysiological conditions that can increase/decrease the mitotic index and can alter or (re)-activate telomerase activity [18]. But, until now, only few studies have explored the relationship between telomere length in different cell types, and the results are unclear [18]. Ferlin et al. [21] showed that telomere length in sperm was strongly correlated with telomere length in leukocytes, although the latter were significantly shorter than in germ cells, which makes sense given the significant telomerase activity in spermatogonia.

The following literature review evaluates the potential link between TL in sperm and leukocytes and male (in)fertility.

## 2. Materials and Methods

This systematic review has been performed following the PRISMA guidelines (the “preferred reporting items for systematic reviews and meta-analyses”) and aims to highlight the link between TL and male fertility. PRISMA is an evidence-based minimum set of items for reporting in systematic reviews. A systematic search of the PubMed database has been performed on the date of 1 December 2022 to identify all human experimental studies of the published literature in English or French (Figure 1). In order to obtain the most complete research, the following keywords have been used: [“telomere length” OR telomerase] AND [“infertile men” OR “male infertility” OR sperm OR semen].

Two hundred and ten articles have been identified. The titles were screened by two independent researchers to evaluate their adequacy. Sixty-two preselected articles were full-text examined, and 22 studies were included in the quantitative synthesis.

For each included study, the following information has been extracted and gathered in tables: authors, year of publication, data on publication, sample, selection criteria, study objectives, number of subjects, age, methods and analysis, study outcomes and main conclusions.

## 3. Results

### 3.1. Subjects

Characteristics of the 3168 participants in the 22 studies and criteria of patients’ inclusion are presented in Table 1.

### 3.2. Telomere Length and Semen Parameters

Numerous interesting studies focused on TL analysis in sperm, and most of them highlighted that short STL is associated with altered semen parameters. All the results are summarized in Table 2. 

One of the most consistent findings is the significant association between semen parameters and STL. Particularly, short telomeres might reflect abnormalities in seminal parameters (count and motility), and might be used as biomarker of qualitatively and quantitatively abnormal spermatogenesis, and therefore would be associated with impaired fertility [21,24,25,26,27,29,31,32,33,38,41]. Eight studies highlighted a correlation between STL and sperm count [13,21,24,25,26,27,32,38,41], two reported a correlation with sperm mobility [29,32] and one with sperm vitality [29]. Only four studies found no association between STL and sperm parameters [23,24,37,39]. 

In leukocytes, LTL also seems to be correlated with semen parameters. Four studies comforted this hypothesis [31,33,38,41], whereas just one showed no significant association [21].

### 3.3. Telomere Length and Fertility

Studies suggesting that STL plays an important role in global fertility, and their results are summarized in Table 3. Eight studies found longer telomere in fertile men’s sperm than in infertile men’s sperm [22,27,28,30,32,34,36,40]. However, five studies found no significant correlation between STL and fertility [13,23,25,35,37].

In leukocytes, only two studies reported by the same authors found a correlation between LTL and fertility [34,36].

Concerning the association between STL and ART results, the data are controversial. Two studies suggested that STL is positively correlated with fertilization rate [34,35] and one find an association between STL and embryo morphology [25]. However, two other studies found no influence of STL on any ART parameters [13,37].

## 4. Discussion

The maintenance of genome integrity is intimately linked to TL [42]; when telomeres are too short, cells can no more divide, and undergo DNA repair errors and finally apoptosis [43,44]. Previous studies have demonstrated that chromosomal abnormalities due to meiotic nondisjunction or DNA damage are more frequent in sub-fertile men compared to general population [45,46,47]. This is the reason that many authors hypothesized that impaired spermatogenesis and sperm parameters could be correlated with short TL. Our findings agree with previously published data; most of the studies demonstrated a significant association between short STL and defective sperm parameters. Oligozoospermic and infertile men have shorter telomeres than the normozoospermic and fertile men [21,24,25,26,27,29,31,32,33,38,41]. Although STL is positively correlated with progressive motility and vitality, the results remain discordant about the association between STL and spermatozoa morphology [24]. 

Assisted reproductive technology (ART) requires sperm preparation to isolate highly motile sperm [48]. Although the swim-up technique is the easiest method [49], some studies reported that this procedure may increase the level of reactive oxygen species [ROS] and fragment the sperm DNA [48]. Nowadays, the most commonly used technique is density gradient centrifugation (DGC) because of its higher selection ability compared with the swim-up method [50]. The TL of selected spermatozoa has been evaluated after both techniques: Yang et al. [26] and Zao et al. [51] found a longer STL in selected sperm compared to sperm from raw semen, but with no superiority of one or the other technique. In contrast, Lafuente et al. found no significant difference in TL before and after sperm selection by either the swim-up or DGC procedure [32]. This conclusion has been confirmed by Lopes et al. [37], combining a swim-up after a density gradient. Finally, further studies are necessary to conclude if the technique of selection can enrich sperm select in sperm with long TL. In addition, it seems important to distinguish in published studies the precise nature of the sperm sample analyzed (raw sperm or selected sample, and which selection technique) [26,51]. 

The TL measurement method is also a huge point to compare the different studies. Indeed, each method has advantages and inconveniences. Southern blot or telomere restriction fragment analysis (TRF) is considered the gold standard for telomere measurement. This technique is based on telomere sequence knowledge and the use of a labeled probe (TTAGGG) to measure the intensity of telomeres on a Southern membrane, and then determine their average length [52]. However, none of the included studies used this technique, probably because it is very time consuming. Telomeres can also be measured with techniques using Q-FISH, which determines the fluorescence intensity of telomeres after hybridization with a fluorescent probe. Due to the hybridization limit of the probes, Q-FISH does not detect fluorescent signals from telomeres that have a repeat number below the hybridization threshold of the probe. Conversely, very intense signals in FISH could represent telomere clusters, but quantification does not allow to differenciate them. Another disadvantage of this technique is that the probe can bind to certain interstitial telomeric sequences located away from the telomeres, thus generating false positives. However, Q FISH method is considerate acceptable [52]. Interestingly, it must be noted that most of included studies were conducted with a qPCR-based method to measure TL. By measuring the telomere signal (T) against a reference signal of a single copy gene (S), the T/S ratio is obtained. This ratio is proportional to the average telomere length and therefore makes it possible to determine the relative length of the telomeres [52]. Q-PCR is relatively easy to perform and does not require a large amount of DNA (about 50 ng). It can be applied to high throughput analysis and is therefore widely used in large populations studies [52]. However, this technique provides only relative quantification and data are not presented in absolute values (in kb). There is also a high variability (≈10%), strongly limiting the results comparison between different laboratories [52]. This method is known to have high measurement error and thus requires a large sample size to offset the error, which was not the case for most of the reviewed studies (less than 100 subjects in almost all).

The ROS level in the spermatozoon of infertile men is significantly higher than in fertile men [53,54]. Several studies demonstrated a link between telomere shortening and metabolic or inflammatory chronic diseases such as diabetes, hypertension or atherosclerosis, which are characterized by systemic oxidative stress [55]. 

In addition, an unhealthy lifestyle and behavioral habits such as smoking, alcohol consumption, nutrition and obesity can increase ROS production and negatively affect the STL [22,56,57,58,59]. In this study, no information regarding the possible exposure of men to ROS was available. However, in their study, Mishra et al. found longer telomeres in the infertile men with mild oxidative stress [28]. Although severe oxidative stress leads to extensive damage to biomolecules, a moderate oxidative stress level could be necessary for STL maintenance and beneficial to cellular homeostasis [60]. 

Telomere length is maintained by telomerase activity. Thus, a telomerase defect could be responsible of shorter telomeres in the germ cell line, reducing sperm production, with repercussions for sperm parameters. This hypothesis agrees with murine experimentation: the fertilization rate and blastocyst formation rate are lower when using sperm from mouse with null telomerase (TR−/−) [42]. Yan et al. identified genetic variants (SNPs) in telomerase reverse transcriptase which are significantly associated with human male infertility [61]. These results were not confirmed by Biron et al. in 2018 [30]. Despite the different germinal cells levels of telomerase activity, Gentiluomo et al. highlighted that the genetic variants TERT-rs2736100 and TERC-rs10936599 are associated with a specific regulation of telomerase [39]. 

Although numerous studies demonstrated an association between short STL and idiopathic male infertility, these same studies have also demonstrated contradictory results [22]. Turner and Hartshorne and others [21,23,35] found no association between STL and semen parameters. This could be explained by the modest samples’ sizes and the variation of parameters in ejaculated semen.

When analyzing the effects of STL on fertility, the use of the cumulative pregnancy rate rather than semen parameters would be more appropriate. Concerning global fertility, sperm with shorter TL seem to be associated with lower pregnancy rates compared with the sperm of men who achieved pregnancy naturally, who had longer STL [22,27,28,30,32,34,36,40]. These data are confirmed by Cariati et al. who showed that no ongoing pregnancies were observed from patients with shorter STL [27], and that pregnancy outcomes tend to be increased when STL is higher.

Furthermore, telomere length in offspring is co-determined by telomere length in parents [62,63]. Consequently, if STL is longer, the embryo will have longer telomeres and a higher cell division ability to achieve pregnancy and live birth. Furthermore, long inherited TL prevents the risk of onset of degenerative disease (Figure 2). This suggests that a man with short sperm telomeres could father with a woman with long oocyte telomere length. 

Five studies found contradictory results [64] but all of them used selected sperm for STL analysis, introducing an important bias, since semen preparation selects spermatozoa with longer telomeres [65].

Previous studies suggested that STL plays an important role in early embryonic development. Using sperm with a higher telomere length could result in better quality embryos and therefore a higher pregnancy rate. The pathogenesis of human infertility is complex, and telomeres are among the first actors to participate to male pronuclear formation after oocyte activation. Therefore, it may be possible that sperm with shorter TL led to abnormal cleavage after fertilization occurs, resulting in poor-quality embryos. In IVF candidates, Yang et al. found a significant association between STL, fertilization rates and good-quality embryos [25]. Berneau et al. and Darmishonnejad et al. also found a significant association between STL and fertilization rate [34,35]. However, the results of Lopes et al. showed no relation between STL and ART parameters such as fertilization rate, good quality embryo rate and blastulation rate. But the reliability of these results is debatable, insofar as the authors did not find any variation of the STL with the age of the patients. In the same way, Torra-Massana et al. didn’t observe any association between STL and fertilization rate or embryo morphology [13] using only sperm donors with normal semen parameters. Moreover, the 2900 embryos analyzed had been obtained with only 60 donors’ semen samples, with a very low average age of 24 years. These assessments did not take into account additional andrological parameters such as sperm DNA integrity which may have a great impact on sperm quality and sperm fertilization capacity. Other studies with a large number of included patients may be necessary to highlight a significant association. Except the studies of Torra Massana et al., [13] and Darmishonnejad et al., [34,36], all included patients undergo ART treatment and may be infertile. In these studies, there is no “control population”, and the cause of infertility is unknown (idiopathic, female and/or male infertility).

A potential limitation of these studies is due to their numerous biases, among which the too small sample size is recurrent; most studies included a small sample of Caucasian or Asian populations, thereby diminishing their validity. This is consistent with previous reviews that have highlighted a high degree of heterogeneity in the studies’ design (age of included patients, heterogeneous severity of infertility, variability of the outcome) [17,66]. 

Telomere dynamics highly depend on age, and most of the included studies have large range of patients’ age to minimize bias. Balmori et al. found that patients with altered semen parameters didn’t show an STL increase with age, contrary to normozoospermic individuals, suggesting a failure of the mechanisms of TL elongation in infertile patients [41]. Ferlin et al. [21] highlighted that children with older mothers and fathers had longer telomere length, and that TL is directly proportional to parents’ age at the time of pregnancy. In these cases, fertile mothers, despite an advanced age, are probably women with inherited long TL, resulting in long TL in oocytes and Granulosa cells [67,68]. 

Considering all the confounding factors mentioned above that can influence the STL (obesity, paternal age, exposure to ROS, environmental pollution…), the prediction of pregnancy chances through STL analysis seems probably more pertinent for an epidemiological approach than an individual one. Nevertheless, this new biomarker could explain some cases of thus far idiopathic infertility, and could constitute an additional parameter to improve the management of sperm donor programs. 

## 5. Conclusions

As an overall conclusion of this systematic review of the literature, numerous studies have been published on the association of telomere length with male semen parameters and clinical pregnancy, and most of them have clearly highlighted the close connection between telomere homeostasis and human fertility. Particular attention has been paid to STL, which was proposed as a new biomarker for male infertility. Although many authors have highlighted a link between STL and semen parameters or clinical outcomes, some of them found contradictory results. Regarding ART parameters, the correlation between STL and fertilization rate or embryo quality remains unclear, and the actual published studies found contradictory results. So, these conclusions have to be confirmed with more subjects and better defined populations in order to control a maximum of biases. STL could be a reliable marker for impaired semen parameters or for the success rate of fertility treatments. It is also essential to pursue the study of the mechanisms of telomeres’ impact on human reproduction. The increasing evidence of telomeres’ role in human reproduction makes them potential markers of fertility, not only of aging. In the future, sperm telomere length could become a prognostic marker for couples in ART centers, and allow better sperm selection.

## Figures and Tables

**Figure 1 genes-14-00425-f001:**
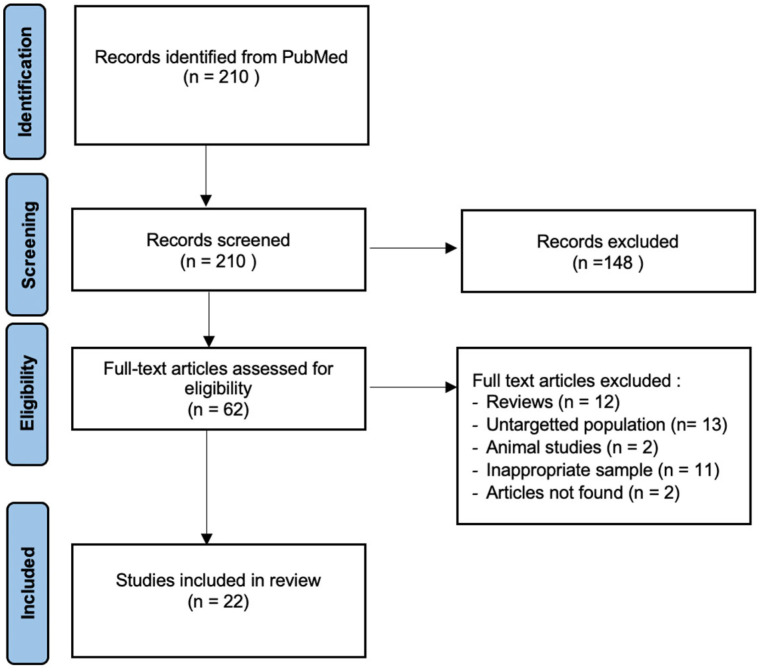
PRISMA flow diagram for identification and selection of studies.

**Figure 2 genes-14-00425-f002:**
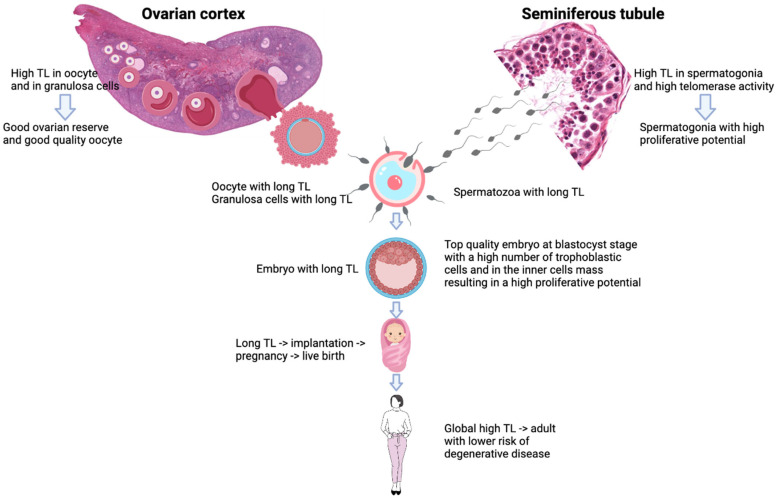
Telomere length inheritance and influences on offspring’s health.

**Table 1 genes-14-00425-t001:** Characteristics of patients and controls in the different studies.

Studies	Number	Criteria of Patient’s Inclusion	Age
Ferlin et al., 2013 [21]	61 normozoospermic men 20 idiopathic oligozoospermic men	Volunteers participating in a screening protocol for prevention of andrological disorders	18–19
Thilagavathi et al., 2013 [22]	32 men with idiopathic infertility 25 controls	Normal female partner examination Men seeking vasectomy and that have fathered a child < 2 years	N.A.
Turner et al., 2013 [23]	45 unselected men	Men undergoing diagnostic semen analysis	21–49
Antunes et al., 2015 [24]	2 oligozoospermic men 1 asthenozoospermic men 1 oligoasthenozoospermic men 6 normozoospermic men	Male partners of couples undergoing ART treatment	32–48
Yang et al., 2015 [25]	418 men undergoing their 1st fresh cycle of IVF	Normal chromosome karyotypes, no Y chromosome microdeletions	30.3 ± 4.0
Yang et al., 2015 [26]	105 infertile men	Infertile men	31.2 ± 6.1
Cariati et al., 2016 [27]	19 oligozoospermic men 54 controls	Men who had requested sperm DNA fragmentation and aneuploidy analyses Normozoospermic men	31–52
Mishra et al., 2016 [28]	112 infertile men 102 controls	Male partners of couples with primary infertility Healthy volunteers	18–45
Rocca et al., 2016 [29]	100 normozoospermic men	Men referred for semen analysis	34.0 ± 8.6
Biron-Shental et al., 2018 [30]	16 sub-fertile men 10 controls	Men undergoing ICSI with normo-ovulatory partners after 1 year of unprotected intercourse Proven fertility in the last year	29–42
Heidary et al., 2018 [31]	30 idiopathic nonobstructive azoospermia 30 controls	Men undergoing ICSI Healthy fertile males	35.4 ± 4.2
Lafuente et al., 2018 [32]	30 infertile patients	Patients attending the fertility clinic for diagnosis	N.A.
Torra Massana et al., 2018 [13]	60 samples used in a total of 676 ICSI cycles	Donor samples used for ICSI	24.3 ± 5
Yang et al., 2018 [33]	247 obstructive azoospermia 349 nonobstructive azoospermia 270 controls	Physical obstruction in the male reproductive system, normal testicular volume, hormone levels, indurated epididymis Spermatogenic dysfunction, abnormal hormone levels, soft testes Sperm counts ≥ 39 × 10^6^/mL, couples with known female factor infertility	25–38
Darmishonnejad et al., 2019 [34]	10 infertile men with previous failed/low fertilization 10 controls	Male partners of couples with previous low [<20%] fertilization rates following ICSI within the last 12 months Couples with ≥ 2 live children of the same sex who requested preimplantation genetic testing for family balancing, percentage of fertilization between 50% and 100%	38.10 ± 4.17 40.11 ± 3.14
Berneau et al., 2020 [35]	65 normozoospermic men	Male partners of couples undergoing ART treatment	25–45
Darmishonnejad et al., 2020 [36]	38 infertile men 19 controls	Infertile men with primary infertility Men with ≥ 1 health child	≤45 20–50
Lopes et al., 2020 [37]	73 unselected men 61 patients	Men undergoing ART treatment Patients undergoing embryo transfer cycles	39.3 ± 4.1
Amirzadegan et al., 2021 [38]	10 oligozoospermic men 10 controls	Sperm count <15 million/mL	40.30 ± 3.75 35.46 ± 5.59
Gentiluomo et al., 2021 [39]	585 unselected men	Men undergoing semen evaluation	18–59
Rocca et al., 2021 [40]	4 oligoasthenoteratozoospermic men 31 normozoospermic men 30 controls	Men from couples who underwent their first fresh ICSI treatment Male partners of couples with successful pregnancy within the first 12 months of regular unprotected sexual intercourse	39 ± 6.4 36.1 ± 6.8
Balmori et al., 2022 [41]	20 normozoospermic men ≤ 25 years 17 oligoasthenozoospermic men ≤ 25 years 20 normozoospermic men ≥ 40 years 20 oligoasthenozoospermic men ≥ 40 years	Sperm donors ≤ 25 years Men undergoing ART treatment ≥ 40 years	21.20 ± 2.35 21.44 ± 2.28 43.30 ± 3.43 43.60 ± 3.95

ICSI, intracytoplasmic sperm injection; N.A., not available; ART, assisted reproductive technology; IVF, in vitro fertilization. Age is expressed as mean ± standard deviation of the mean or as an age range.

**Table 2 genes-14-00425-t002:** Results for telomere length and sperm parameters.

Studies	Samples	Method of Telomere Measurement	Results—Main Findings	*p*
Ferlin et al., 2013 [21]	Sperm et leukocytes	qPCR	Significant positive correlation between STL and sperm count: rS = 0.33	0.0029 *
			No significant positive correlation between LTL and sperm count: rS = 0.003	0.9780
			Oligozoospermic men: STL = 0.95 ± 0.22	0.0001 *
			Normozoospermic men: STL = 1.24 ± 0.25	
			No difference was observed in LTL between the 2 groups	
Turner et al., 2013 [23]	Sperm	Q-FISH	No association between semen parameters or male fertility and STL	>0.05
Antunes et al., 2015 [24]	Sperm	qPCR	Normal semen parameters: STL = 16.63 ± 22.29	0.0001 *
			Abnormal semen parameters: STL = 6.92 ± 18.13	
			Morphologically normal spermatozoa: STL = 12.40 ± 19.77	0.991
			Morphologically abnormal spermatozoa: STL = 13.11 ± 22.56	
Yang et al., 2015 [26]	Sperm	qPCR	Significant positive correlation between STL and total sperm number: r_p_ = 0.53	<0.01 *
Yang et al., 2015 [25]	Sperm	qPCR	Significant correlation between STL and sperm count: r_p_= 0.28	0.001 *
Cariati et al., 2016 [27]	Sperm	qPCR	Normozoospermic group: STL = 1.4 ± 0.1	0.0024 *
			Oligozoospermic group: STL = 0.9 ± 0.1	
			Significant positive correlation between STL and sperm count: r = 0.325	0.006 *
Rocca et al., 2016 [29]	Sperm	qPCR	STL is positively associated with:	
			Progressive motility rp = 0.46	0.004 *
			Sperm vitality rp = 0.340	0.007 *
Heidary et al., 2018 [31]	Leukocytes	qPCR	Azoospermic men: LTL = 0.54 Fertile men: LTL = 0.84	<0.05 *
Lafuente et al., 2018 [32]	Sperm	Q-FISH	Significant positive correlations between:	
			STL and sperm concentration: r = −0.308	0.049 *
			STL and progressive motility: r = −0.353	0.028 *
			STL and immotile sperm: r = 0.446	0.007 *
			No significant correlation between STL and sperm DNA fragmentation	>0.05
Torra-Massana et al., 2018 [13]	Sperm	qPCR	No relevant correlation between STL and sperm motility	0.34
			Significant correlation between STL and sperm concentration	0.03 *
Yang et al., 2018 [33]	Leukocytes	qPCR	Controls + OA: LTL = 0.96; NOA: LTL = 0,81 [OR 0.172; 95% CI 0.107–0.279] No significant difference between OA and normozoospermic controls	<0.001 *
Lopes et al., 2020 [37]	Selected sperm	qPCR	No relation between STL and total sperm count	0.590
			No relation between STL and sperm motility	0.354
			No relation between STL and normal morphology	0.169
Amirzadegan et al., 2021 [38]	Leukocytes and sperm	qPCR	Oligozoospermic men: LTL = 0.61 ± 0.30	0.01 *
			Fertile men: LTL = 1.22 ± 0.71	
			Oligozoospermic men: STL = 0.65 ± 0.25	0.02 *
			Fertile men: STL = 1.04 ± 0.46	
Gentiluomo et al., 2021 [39]	Sperm	qPCR	No significant association between STL and sperm parameters	>0.05
Balmori et al., 2022 [41]	Sperm and PBMC	Q-FISH	In NZ ≤ 25 years, significant positive correlation between STL and	
			Sperm count: r = 0.641	0.009 *
			Motility: r = 0.639	0.007 *
			OAZ men ≤ 25 years: TL shorter than NZ men ≤ 25 years; NZ men ≥ 40 years;	0.0081 *
			OAZ men ≥ 40 years	0.0116 *
				0.009 *

PBMC, peripheral blood mononuclear cells. Results are expressed as mean STL ± standard deviation of the mean. STL determined by a qPCR-based method could be expressed as absolute or relative mean telomere length or correlation coefficient. * *p* values < 0.05 are significant.

**Table 3 genes-14-00425-t003:** Results for telomere length and fertility.

Studies	Samples	Method of Telomere Measurement	Results—Main Findings	*p*
Clinical criteria of fertility				
Thilagavathi et al., 2013 [22]	Sperm	qPCR	Infertile men: STL = 0.674 ± 0.028	<0.005 *
			Controls: STL = 0.699 ± 0.030	
Turner et al., 2013 [23]	Sperm	Q-FISH	No association between male fertility and STL	>0.05
Yang et al., 2015 [25]	Selected sperm	qPCR	No significant association between STL and clinical pregnancy rates	0.90
Cariati et al., 2016 [27]	Sperm	qPCR	STL between 0.2–2.0: pregnancy rate = 35.7%	0.04 *
			STL < 0.2 or > 2.0: pregnancy rate = 0.0%	
Mishra et al., 2016 [28]	Sperm	qPCR	Infertile men: STL = 0.609 ± 0.15Controls: STL = 0.789 ± 0.060	<0.0001 *
Biron-Shental et al., 2018 [30]	Sperm	FISH	Sub-fertile sperm: STL = 0.6 ± 1.2%Fertile sperm: STL = 3.3 ± 3.1%	<0.005 *
Lafuente et al., 2018 [32]	Selected sperm	Q-FISH	Previously achieved a natural clinical pregnancy STL = 26.17 ± 8.20 kb	0.024 *
			Couples who had never conceived STL = 19.50 ± 5.05 kb	
Torra-Massana et al., 2018 [13]	Thawed and selected sperm	qPCR	No significant effect of STL on reproductive outcomes:	
			Biochemical pregnancy rate	0.411
			Clinical pregnancy rate	0.986
			Ongoing pregnancy rate	0.769
			Live birth rate	0.595
Darmishonnejad et al., 2019 [34]	Sperm and leukocytes	qPCR	Infertile men: STL = 0.74 ± 0.15	<0.05 *
			Fertile men: STL = 1.24 ± 0.18	
			Relative telomere length in leukocytes was no different between the two groups	
Berneau et al., 2020 [35]	Selected sperm	qPCR	Clinical pregnancy: STL = 1.026 ± 0.013	0.188
			No clinical pregnancy: STL = 0.995 ± 0.016	
Darmishonnejad et al., 2020 [36]	Sperm and leukocytes	qPCR	Fertile men: STL = 1.09 ± 0.13	0.002 *
			Infertile men: STL = 0.61 ± 0.07	
			Fertile men: LTL = 1.1 ± 0.14	0.1
			Infertile men: LTL = 0.76 ± 0.07	
Rocca et al., 2021 [40]	Sperm	qPCR	ART group: STL = 0.9 ± 0.3	0.02 *
			Control group: STL = 1.2 ± 0.6	
Lopes et al., 2020 [37]	Selected sperm	qPCR	No relation between STL and time of infertility	0.556
			No influence of relative STL on implantation rate	0.508
ART embryo criteria				
Yang et al., 2015 [25]	Selected sperm	qPCR	STL is positively associated with the good embryo quality rate	<0.001 *
			STL is positively associated with the transplantable embryo rate	<0.001 *
			No association between STL and fertilization rate	0.49
Torra-Massana et al., 2018 [13]	Thawed and selected sperm	qPCR	No significant correlation between STL and the average score of embryo morphology	0.08
			No significant correlation between STL and the fertilization rate	0.35
Darmishonneiad et al., 2019 [34]	Sperm	qPCR	Positive significant correlation between fertilization rate and STL	0.007 *
Berneau et al., 2020 [35]	Selected sperm	qPCR	STL is positively correlated with fertilization rate	0.004 *
			No significant association between STL and embryo cleavage rate	>0.05
Lopes et al., 2020 [37]	Selected sperm	qPCR	No influence of relative STL in fertilization rate	0.411
			No influence of relative STL on embryo cleavage rate	0.900
			No influence of relative STL on AB embryo grade rate	0.123
			No influence of relative STL on embryo fragmentation	0.136
			No influence of relative STL on blastocyst formation rate	0.836

Results are expressed as mean STL ± standard deviation of the mean. STL determined by a qPCR-based method could be expressed as absolute or relative mean telomere length or correlation coefficient. * *p* values < 0.05 are significant.

## Data Availability

Not applicable.

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
