# Peer review of "Telomere Length, a New Biomarker of Male (in)Fertility? A Systematic Review of the Literature"

_genes, 2023, doi:10.3390/genes14020425_

Round 1

Reviewer 1 Report

Dear Editor and Authors,

I reviewed the paper entitled “Sperm telomere length and telomerase activity as new biomarkers for male infertility? A systematic review of the literature” submitted for publication on “Genes”.

The main conclusions indicate that shorter sperm telomeres appear to be associated with altered semen parameters or male infertility. Telomere length may be considered as a new molecular marker of sperm quality and male fertility potential.

Overall, the take home message is scientifically sound. Nonetheless, I found that some relevant information is not properly classified or revised:

One of the most relevant data extracted from the literature could be the potential relation of sperm telomers length with fertility, as is indicated in the title and in the abstract (last lines). These possible relations have been discussed (lines 234-260). However, is the available information sufficient to summarize the most relevant fertility results in a new table? The idea is to summarize the information linking the fertility data and sperm telomeres length in a new table (pregnancy rates, fertility rate and embryo quality in ART,…)

Another point that deserves further description is the possible relation between donors age and sperm telomeres length. It should be further discussed.

Minor changes that could improve the manuscript are:

It should be further explained in the introduction or in the discussion why are included leukocyte data. Are leukocytes telomeres length studied to compare with sperm cells? What are the implications of leukocytes telomers length in relation to human reproduction?

It should be revised the reference list format since the publication dates are noted in French.

Reviewer 2 Report

In this review article, Fattet et al provided a systematic review of literature on sperm telomere length and telomerase activity as biomarkers for male infertility. There is a strong interest in finding new parameters/biomarkers for the detection of sperm abnormalities that are associated with male infertility. However, because of significant genetic variations in telomere length or telomerase activity, together with the inaccuracy of telomere measurement and telomerase detection, telomere length is unlikely to be a parameter with strong association with male infertility. Nevertheless, the authors reviewed 22 publications that included more than 4000 subjects. The authors found that some of the studies showed longer telomere length in fertile men, but a significant number of publications reported conflicting results. Therefore, while this review provided useful information on this subject, its overall impact on clinical applications of telomere length and telomerase measurement in male fertility evaluation is limited. Below are some specific comments.

1.     Explain ART and PRISMA in the beginning (abstract) of the article.

2.     For measuring telomere length, TRF Southern is the gold standard, and flow FISH/quantitative FISH with appropriate controls are acceptable. However, quotation about sperm telomere length is misleading. The authors should analyze the original data, emphasizing using the acceptable methods, especially Southern TRF data.

3.     Table 3, telomerase: method of detection and quantification for each publication are not listed. Some of refs in this table did not measure telomerase activity at all.

4.     In all table 1, 2, and 3, the authors should analyze age of subjects, methods of sample collection, as well as methods of quantification, because all these procedures are likely impact the experimental data.

Reviewer 3 Report

In the peer-reviewed systematic review, the authors analyze the possibility of using sperm telomere length and telomerase activity as new biomarkers for male infertility. The authors conclude that there is an association between short sperm telomeres and altered semen parameters. Data on the existence of a relationship between male infertility and telomerase activity in sperm or telomere length in leukocytes is controversial according to the authors. Although data on the relationship between a decrease in telomere length and impaired cellular functions cannot be considered new, systematization of data in this area may be useful, given the relevance of the problem of searching for new markers of infertility. However, in my opinion, the article needs to be improved.

Major point

Serious questions are raised by the section “Telomerase activity” and, in particular, table 3. The table is titled “Results for telomerase activity”. However, none of the refereed studies specifically assessed telomerase activity. In addition, some wording in the table (column Results - Main findings), in my opinion, does not clearly reflect the results of the analyzed studies.

Biron-Shental et al. 2018 [26]: authors of the MS write about “expression of telomerase”. In the original paper, it is talking about “The percentage of cells that were positive for hTERT”. This is not the same.

Reig-Viader et al. 2014 [27]: authors of the MS write that sperm samples were studied. In the original paper, it is talking about testicular biopsies, not sperm. Detection of telomeric repeat-containing RNA (TERRA) and telomerase distribution in testis cell spreads by combining immunofluorescence and RNA fluorescent in situ hybridization cannot be considered as a direct indicator of telomerase activity.

Given that the authors have analyzed only 5 studies in this section and none of them are directly assessed telomerase activity (for example, using the ELISA method), I can recommend the authors to exclude this section from MS. Accordingly, in this case, correction of the MS header is also required.

Minor point

It would be useful to discuss cases of telomere length reduction in men with normal spermogram and normal fertility, if any are described in the literature.

Round 2

Reviewer 2 Report

The authors have addressed my concerns.